# Effects of Aging on Hair Color, Melanosomes, and Melanin Composition in Japanese Males and Their Sex Differences

**DOI:** 10.3390/ijms232214459

**Published:** 2022-11-21

**Authors:** Takashi Itou, Shosuke Ito, Kazumasa Wakamatsu

**Affiliations:** 1Kao Corporation, R&D—Hair Care Products Research, Tokyo 131-8501, Japan; 2Institute for Melanin Chemistry, Fujita Health University, Toyoake 470-1192, Japan

**Keywords:** male hair, aging, melanosome, melanin, sex difference, pheomelanin, 5,6-dihydroxyindole

## Abstract

In a previous study, we observed that the hair color of Japanese females darkens with age and that the causes of this are the increase in melanosome size, the amount of melanin, and the mol% of 5,6-dihydroxyindole (DHI) which has a high absorbance. In this study, we extended the same analyses to male hair to examine the sex differences in hair color, melanin composition, and melanosome morphology. Male hair also tended to darken with age, but it was darker than female hair in those of younger ages. Although there was no age dependence of DHI mol% in male hair, as with female hair, the melanosomes’ sizes enlarged with age, the total melanin amount increased, and these findings were correlated with hair color. The analyses, considering age dependence, revealed that there were significant sex differences in the ratio of absorbance of dissolved melanin at the wavelength of 650 nm to 500 nm, in pheomelanin mol%, and in melanosome morphology parameters such as the minor axis. This may be the cause of the sex differences in hair color. Furthermore, the factors related to hair color were analyzed using all the data of the male and female hairs. The results suggested that total melanin amount, pheomelanin mol%, and DHI mol% correlated with hair color.

## 1. Introduction

Hair has a role in protecting the head and excreting harmful elements, and it has an impact on a person’s appearance. The influence of aging on hair is known to include changes in thickness, density, shape, and hair color [1,2,3]. Most studies on these factors used female hairs. For male hairs, there are many studies on hair thinning [4,5], but little is known about the sex differences in the physical properties of hair; though there are studies on the trace elements in hair [6,7], hair thickness [8], and hair color during the period growth [9,10].

In the case of the pigmented hairs in Asia, such as those in Japanese people, hair color is black–brown and becomes gray due to aging. This is because melanin, being the source of hair color, is no longer generated. There have been many studies on the cause of graying hair [11,12]. Before graying, the color of pigmented hair may change with age. It is generally known that, in the Japanese population, the pigmented hair color is brown during childhood but that it turns darker as children grow. Recently, in a study on Japanese females, we observed that individuals’ hair colors continued to turn darker with age after pubertal age [13,14]. However, less is known about hair color changes in males.

Melanin is produced in the melanocytes in hair follicles [15]. There are two types of melanin: black to brown eumelanin and yellow to reddish-brown pheomelanin. Eumelanin consists of two types of moieties which are derived from 5,6-dihydroxyindole (DHI) and 5,6-dihydroxyindole-2-carboxylic acid (DHICA) [16,17]. Melanosomes containing melanin pigments are produced in the melanocytes in hair follicles, transported through dendritic melanocytes, and incorporated into keratinocytes.

We already analyzed the melanin composition and melanosome morphology in the hair of Japanese females within a wide age range, and we observed that melanosomes enlarged with age, that the amount of melanin increased with age, and that these age dependencies were the causes of the age-dependent darkening of hair color [14]. In addition, the mole percent of DHI moiety in eumelanin (DHI mol%) also increased significantly with age, which may contribute to further darkening because eumelanin derived from DHI had higher absorbances than that derived from DHICA.

The purpose of this study was to extend the study of the age dependence of hair color, melanin composition, and melanosome morphology in female hairs to male hairs to investigate their sex differences as well as to elucidate the factors related to hair color via an analysis using all data regardless of sex.

## 2. Results

All the results obtained in this study are summarized in Appendix A. Including two subjects whose hair-color measurements could not be performed due to the small amounts of their hair samples, the number of subjects was forty-four, ranging from 4 to 72 years old.

### 2.1. Age Dependence of Hair Color

The results of the age dependence of male hair color measured on 42 hair samples are shown in Figure 1. All hair color parameters, *L**, *a**, and *b**, significantly decreased with age (Table 1). Although the data varied widely, the results showed that the trend was qualitatively similar to that in female hairs [14]; that is, hair color tended to darken from childhood to adulthood.

### 2.2. Melanin Compositions

Melanin analyses were performed on 44 male hairs in almost the same manner as the procedures described in the previous paper [14]. The results are summarized in Appendix A, and the age correlations are shown in Figure 2. The results in the previous paper on female hairs were also included for comparison. Figure 2a,b show the total melanin amount (TM) obtained from the absorbance at 500 nm of hair melanin that was solubilized in Soluene-350 and the absorbance ratio of 650 nm to 500 nm, A650/A500, respectively. A high positive correlation of the TM with age was observed (*p* = 4.6 × 10^−6^). A650/A500 also had a significant positive correlation with age (*p* = 0.016) although it was less correlated than in the case of the female hairs. From the above information, it was suggested that the color of solubilized melanin changed to darker black with age.

Figure 2c–e show the age dependencies of pyrrole-2,3,5-tricarboxylic acid (PTCA), pyrrole-2,3-dicarboxylic acid (PDCA), and pyrrole-2,3,4,5-tetracarboxylic acid (PTeCA), respectively, which are markers of the eumelanin component obtained by high-performance liquid chromatography (HPLC) after decomposing hair eumelanin through alkaline hydrogen peroxide oxidation (AHPO). The plots of the male hairs nearly overlapped those of the female hairs and were significantly correlated with age (PTCA *p* = 0.00023, PDCA: *p* = 0.00067, PTeCA: *p* = 0.0095). On the other hand, as shown in Figure 2f, the ratio of PTeCA to PDCA had a negative and significant correlation with age in the female hairs, whereas an age correlation was not observed in the male hairs.

The age dependence of the DHI mol%, determined from the ratio of PDCA/PTCA, is shown in Figure 2g, and that of the pheomelanin mol%, obtained from the ratio of thiazole-4,5-dicarboxylic acid (TDCA) to PDCA which was evaluated via the acid hydrolytic treatment of the hair by HCl followed by AHPO and HPLC analyses [18], is shown in Figure 2h. In male hairs, no age dependence was observed in either plot.

### 2.3. Melanosome Morphology

We enzymatically isolated melanosomes from the male hairs and observed them using a scanning electron microscope (SEM). Representative SEM images of the isolated melanosomes are shown in Appendix A. Their major and minor axes were measured, and the aspect ratio (major/minor axes) and the volume of the melanosomes were calculated from these axes. Along with the results of the female hairs [13,14], the mean values of these parameters were plotted against age in Figure 3. In the female hairs, the major axis (Figure 3a) was not age-dependent, but in the male hairs, the major axis tended to be slightly longer with age (*p* = 0.021). Next, we could see a tendency to increase markedly with age in the minor axis (Figure 3b) (*p* = 7.8 × 10^−6^). The histogram of the minor axis for each age group is shown in Appendix A. The distribution of the minor axis was the narrowest for the hair of children under 10 years old, but the distribution expanded with age. Additionally, melanosomes with a minor axis of 0.6 μm or more were clearly observed over 50 s. Such a spread of the minor axis distribution was also found in the female hairs [13].

The aspect ratio (Figure 3c) had a significant negative correlation with age (*p* = 0.0017), meaning that the shape of the melanosomes became thicker with age. The slope against age for the male hairs was relatively mild compared with the female hairs, and the divergence of the aspect ratio between the male and female hairs was larger for younger ages. It was observed that the shape of the melanosomes in the young female hairs comprised slender ellipsoids and was clearly different from that of the adults [13,14]; however, the shape of the melanosomes in the young male hairs was rather difficult to distinguish from that of the adults. The mean volume of the melanosomes *V* was estimated from the major and minor axes with an assumption of ellipsoids. The age correlation of the *V* was very high (*p* = 3.9 × 10^−5^), and it was also confirmed in the male hairs that the melanosomes’ volume significantly increased with age (Figure 3d).

We examined the correlations between the values determined by the melanin analyses and the *V* in Appendix A. It was confirmed in the male hairs that the TM became significantly higher as the *V* became larger (*p* = 0.00023). On the other hand, no correlation of the *V* was found with A650/A500, the DHI mol%, and the pheomelanin mol%. These results were similar to those observed in the female hairs [14].

## 3. Discussion

### 3.1. Causes of Hair Color Change with Age in Male Hair

In order to examine the cause of the hair color change, the *L**, *a**, and *b** values of the male hairs were plotted against various parameters obtained by the melanin analyses in Figure 4, and the *p*-values of the correlations are summarized in Table 1. It was observed that *L** had the highest correlation with the PTCA levels (*R*^2^ = 0.27, *p* = 0.00041). PTCA levels are markers corresponding to the amount of DHICA in eumelanin, which accounts for the majority of melanin [19]. In addition, the correlation of *L** with the TM (Figure 4a) (*R*^2^ = 0.22, *p* = 0.0016) was higher than that with age (Figure 1) (*R*^2^ = 0.19, *p* = 0.0035). This is reasonable because the TM is the parameter that reflects hair color more directly, while age dependence includes a variation due to individual differences. The same tendency was shown in the female hairs [14]. The A650/A500 ratio did not significantly correlate with *L**, *a**, and *b** (Figure 4b). Neither the DHI mol% nor the pheomelanin mol% correlated with *L**, but significant correlations with the *a** and *b** values were observed (Figure 4c,d), suggesting that both may contribute to hair color changes. On the other hand, the PDCA level that was related to the DHI unit was significantly correlated with *L**, and high correlations with *a** and *b** were seen (Table 1).

Next, the relationships between hair color and the melanosome morphology parameters are shown in Figure 5, and each *p*-value is summarized in Table 1. Significant correlations at the 5% level were observed for *L** in the major axis, minor axis, and *V*, and, for *a** and *b**, significant correlations were observed in the minor axis, aspect ratio, and *V*. The major axis, which had a slight correlation with age, also had a weak correlation with *L** (Figure 5a). For the aspect ratio, very high correlations with *L**, *a**, and *b** were observed in the female hairs [14] whereas, in the male hairs, there was no or a weak correlation (Figure 5c). The melanosome volume (Figure 5d) was significantly correlated with *L**, *a**, and *b**, indicating that the hair color darkened as the size of the melanosomes increased. With the correlation of the TM with the *V* (Appendix A), it could be said that the hair color darkened as the TM increased as a consequence of the enlargement of the *V* with age.

### 3.2. Sex Differences

We discussed the sex differences in the various parameters obtained in this study. For the female hairs, the number of data was twenty-five for hair color, thirty-three for melanin composition, and thirty-eight for melanosome morphology [14].

#### 3.2.1. Hair Color

Figure 6 shows comparisons of the hair color parameters between the male and female hairs. Since the plot of *b** vs. that of *a** for both types of hairs overlapped and followed a straight line through the origin (Figure 6a), it can be said that the hues were not different between the male and female hairs, but, overall, the male hairs had a smaller *L** (Figure 6b), meaning that they had a darker hair color than the female hairs. The difference in *L** was particularly large during young age periods. When the *L** values were analyzed with the analysis of variance (ANOVA) in four categories of age (20 years old and under, 21–40, 41–59, and more than 60 years old) and the sex difference of *L** was examined, the *p*-values in these age periods were 0.012, 0.016, 0.95, and 1.0, respectively (Appendix A), indicating a significant sex difference in the two groups up to 40 years old.

Previous findings on the sex differences in hair color include studies on Viennese individuals in a wide age range [9] and on Polish individuals aged 7–10 years [10]. Reuer [9] showed that hair color darkened with age and that female hairs were slightly darker at the period of intensified growth because girls develop faster than boys, but there was no significant sex difference in hair color in other age periods. Sitek et al. [10] indicated that girls are characterized by possessing stronger pigmentation than boys in the skin and hair at the stage of ontogeny. In these studies, the sex differences in hair color were considered to have arisen, reflecting a difference in development. The sex differences in hair color observed in this study were not within such a limited age period, and this is the first finding of this phenomenon to the best of our knowledge.

#### 3.2.2. Melanin Compositions

Figure 2 includes the results of 33 female hairs. Although the male hair data slightly shifted to the upper regions in A650/A500, they almost overlapped in the age-dependent plots of the TM, PTCA, PDCA, and PTeCA, and no sex differences were found. In the case of PTeCA/PDCA, an indicator of the proportion of cross-linked and non-cross-linked melanin [20], the plots of the female hairs showed a significant decrease with age [14] whereas there was no correlation with age in the male hairs. It is known that DHI melanin has a more complex aggregation structure than DHICA melanin [21] and that it may be less likely to be cross-linked by heat. It was considered reasonable for the female hair that the proportion of cross-linked melanin decreased with age since the ratio of DHI increased with age [14]. On the contrary, in the case of the male hair in this study, since the DHI mol% had no correlations with age (Figure 2g), it could be considered that the proportion of cross-linked melanin also did not show any correlations with age, which is consistent with the discussion in the previous paper [14].

We examined the sex differences in the TM, A650/A500, the DHI mol%, and the pheomelanin mol%. If the regression lines against age for the male and female hairs were parallel with each other, the sex difference was analyzed using the analysis of covariance (ANCOVA), and if they were not parallel, it was analyzed using the ANOVA. The results are summarized in Table 2. As a result, we found sex differences at highly significant levels in the pheomelanin mol% (*p* = 5.3 × 10^−7^) and a significant difference in A650/A500 (*p* = 0.0058).

Melanin biosynthesis begins with L-tyrosine, the starting material, and is oxidized with tyrosinase to dopaquinone. It is followed by two pathways [17]. In one pathway, dopaquinone undergoes intramolecular cyclization to form cyclodopa, and it finally synthesizes eumelanin. In the other pathway dopaquinone reacts with cysteine to form 5-*S*-cysteinyldopa followed by a route to pheomelanin synthesis. Due to the rapid reaction of dopaquinone with cysteine, it is thought that pheomelanin is first produced and that eumelanin is synthesized after most cysteine is consumed [16,17]. The TM and melanosome size increased with age, but the pheomelanin mol% had no correlation with age, which means that the molar ratio of cysteine in the melanocytes is independent of the TM and the size of melanosome. Conversely, the male hair had a lower pheomelanin mol%, suggesting that the male hairs had a lower rate of cysteine in the melanocytes than the female hairs had. However, there is no evidence to support this so far, and, on the contrary, there is a report describing that men have significantly higher levels of cysteine in their blood [22]. The cause of the sex difference in the pheomelanin mol% in hair is unknown.

#### 3.2.3. Melanosome Morphology

The sex differences were examined in the same manner as described above with respect to the major axis, minor axis, aspect ratio, and the volume of the melanosomes, and the results are summarized in Table 2. Significant sex differences were observed in these parameters, and these differences were especially remarkable in the minor axis and the volume when considering the effect of age dependence (minor axis: *p* = 6.0 × 10^−9^, volume: *p* = 7.1 × 10^−7^). That is, it could be said that the melanosomes in the hair were larger in males than in females.

A possible reason for the sex differences and age dependence in melanosome size is a difference in the hair growth rate. Melanosomes are made inside melanocytes and are transported to keratinocytes. If the growth of keratin becomes slower, it is expected that the time of delivery becomes longer, and the melanosome’s size could be larger. As for the sex difference in the hair growth rate, a study on Chinese individuals was reported by Liu et al. [23] where the hair growth rate was measured for 20 males and 21 females with an average age of 30 years, and it was found that hair growth was slightly faster for females. Miyamoto et al. [24] reported a quantitative study on hair growth rates in Japanese males using a phototrichogram, which compares an enlarged image of a small area of the head taken after cutting the hair near the scalp with an enlarged image of the same area taken a few days later, indicating that the hair growth rate decreased from 0.34 mm/day to 0.21 mm/day in the parietal region and from 0.36 mm/day to 0.31 mm/day in the temporal region of the head from the age of 25 to 55. On the other hand, Kidena et al. [25] used the same method for Japanese females and found that the hair growth rate decreased from 0.38 mm/day at the age of 30 s to 0.30 mm/day at 60 s in both the parietal and the temporal regions of head. Furthermore, although the ethnic origin is unknown, Turner et al. [26] also reported a decline in the growth rate of female hairs from 16 μm/h (=0.384 mm/day) before the age of 20 to 14 μm/h (=0.336 mm/day) at the age of 65. These data indicate that the hair growth rate slows down with age in both males and females and that hair growth is slower in males than females. As described above, in the viewpoint of the age dependence and the sex difference, the tendencies of the hair growth rate and melanosome size are not contradictory.

In general, it is considered to be that when the melanosomes’ size is larger, the TM increases (Appendix A), which does not fit the fact that there were no sex differences in the TM. The reason why the melanosomes’ size in the male hair was larger than that in the female hair could be that the number of melanosomes per hair fiber, or melanin density in the melanosome was different. However, accurate measurements are difficult to obtain, and the reason for this phenomenon remains unsolved.

### 3.3. Factors Contributing to Hair Color (Analysis with the Mixed Data Set of Male and Female Hairs)

According to the comparison of hair color between males and females, the male hairs had a darker hair color (Figure 6b), but there were no significant sex differences in the amount of the TM (Figure 2a, Table 2). Regardless of sex, hair color should reflect the melanin composition in the hair. Therefore, we combined all the results of the males and females (n = 67) and explored the parameters that reflected the hair color. Examining the correlations of the hair color parameters *L**, *a**, and *b** with the levels of PTCA, PDCA, PTeCA, thiazole-2,4,5-tricarboxylic acid (TTCA), and 4-amino-3-hydroxyphenylalanine (4-AHP), in the TM, DHI mol%, and pheomelanin mol% for all subjects, we found a weak correlation not only with components relating to eumelanin but also with those related to pheomelanin (Appendix A). In addition, *a** and *b** showed very high correlations with PDCA and with the DHI mol%, which is associated with PDCA.

Setting *L**, *a**, and *b** as the objective variables and setting the TM, DHI mol%, and pheomelanin mol% as the explanatory variables, we performed multiple regression analyses and obtained the following results. Here, in the analysis of *L**, the DHI mol% was excluded because it had no correlations with *L**.
*L** = 18.55 − 0.2215 × TM + 0.1043 × pheomelanin mol%(*R*^2^ = 0.44, *p* = 8.8 × 10^−9^)(1)
*a** = 8.026 − 0.1622 × TM − 0.0522 × DHI mol% + 0.0756 × pheomelanin mol%(*R*^2^ = 0.62, *p* = 3.5 × 10^−13^)(2)
*b** = 10.16 − 0.2115 × TM − 0.0610 × DHI mol% + 0.0926 × pheomelanin mol% (*R*^2^ = 0.58, *p* = 6.6 × 10^−12^)(3)

All parameters exhibited a higher correlation than the case of TM alone (Appendix A). At the 5% level, the pheomelanin mol% was significant for all the hair color parameters, and the DHI mol% was significant for *a** although the *p*-value for *b** was 0.067 (Appendix A). From the above information, it was suggested that the pheomelanin mol% also affected the lightness *L** of hair, and the pheomelanin mol% and the DHI mol% also affected the color’s shade. Looking at the coefficients, the mol% of pheomelanin, which is yellow to reddish-brown and lighter than eumelanin, was indeed positively correlated. On the other hand, the DHI mol% was negatively correlated with red (*a**) and yellow (*b**) (and positively correlated with green and blue). From the measurements of absorbance on the melanin synthesized with various ratios of DHI and DHICA, it was observed that the higher the ratio of DHI, the higher the absorbance in the wavelength of 400–800 nm and the larger the A650/A500 ratio [14]. That is to say that, since DHI-derived melanin absorbs relatively more red color, it is reasonable that the DHI mol% had a negative correlation with *a**. In female hairs, the DHI mol% increased significantly with age, and it was thought that the darkening of hair with age was due to the increase in the DHI mol% in addition to the increase in the TM, while the pheomelanin mol% had no correlation with age. Additionally, a relationship between the pheomelanin mol% and hair color was not found. In this study, however, the pheomelanin mol% in the male hairs was lower than that in the female hairs, and the hair color was darker in the male hairs, which led to a new finding that hair color also correlated with the pheomelanin mol%.

It was reported that the melanosomes isolated from red hairs in which pheomelanin was rich were less regularly shaped; both ellipsoids and spheres could be found, and they were generally smaller than those isolated from black hairs [27]. In the frequency distribution of the aspect ratio of the melanosomes in the male hairs (data not shown), the frequency of an aspect ratio less than 1.5 was almost nil, suggesting that there were almost no melanosomes such as those found in red hairs. It has been proposed from a kinetic point of view that a core of pheomelanin is first formed in a melanosome, and eumelanin forms around it [16], in which case pheomelanin is expected to be buried inside melanosomes. Even so, some light will penetrate the melanosomes, and they will also reflect the color of the pheomelanin inside.

### 3.4. DHI Ratio and Its Relationship to Melanosome Morphology

In both the male and female hairs, the aspect ratio of the melanosomes was larger in younger ages, and it decreased with age (Figure 3c). The sex difference was significant, with the male hairs having a smaller aspect ratio than the female hairs (Table 2). In particular, in the younger period, the sex difference was large; the melanosomes of the female hairs had an elongated and relatively uniform shape [13,14], while in the male hairs, the melanosome shape of the younger period was relatively thick and not substantially different from that of the adults.

In the female hairs, the PDCA levels were highly correlated with the aspect ratio (*R*^2^ = 0.43, *p* = 3.2 × 10^−5^), and the DHI mol% obtained from PDCA/PTCA was also correlated with the aspect ratio (*R*^2^ = 0.21, *p* = 0.007) [14]. Since synthetic DHI melanin formed small, globular aggregations (<100 nm) and synthetic DHICA melanin formed rod-like assemblies (ca. 1 μm) [21], it was thought that the proportion of DHI may have some effect on the morphology of melanosomes. We also discussed the correlation between the aspect ratio and the DHI mol% for the male hairs obtained in this study. The aspect ratio was plotted against the DHI mol% in Figure 7a. A weak correlation was also seen for the male hairs although it was not significant (*R*^2^ = 0.085, *p* = 0.055). In addition, the plots of the male hairs did not overlap those of the female hairs, and they appeared to have a different relationship. In the case that the data of the males and females were combined, the *p*-value decreased due to the increase in the number of subjects (*R*^2^ = 0.11, *p* = 0.0032); consequently, the effect of the DHI mol% on the aspect ratio was further confirmed.

Recently, Ito et al. [28] re-examined the conditions in the HPLC measurement and found that a separation of melanin markers improved by adding tetra-*n*-butylammonium bromide, an ion-pair reagent, to the HPLC elution buffer. As a result of comparing the improved method to the conventional method, it was found, in the case of human hair, that the amount of PTCA, TTCA, and TDCA almost reproduced the results by the conventional method, but that of the amount of PDCA measured with the improved method was slightly lower than that measured with the conventional method. Additionally, the deviation was relatively large in the high concentration region. In this study, the measurements on the male hairs were performed with the improved method and were different from those of the female hairs performed with the conventional method in the previous study. To predict what would have happened if the measurement of female hairs had been performed with the improved method, the previous results of the PDCA levels in female hairs were corrected using the following relational equation described in the paper by Ito et al. [28]:PDCA (improved) = 0.776 × PDCA (original) + 1.81(4)

Furthermore, the corrected DHI mol% was calculated with the values of the corrected PDCA levels and was plotted in Figure 7b. As a result of the correction, the correlation between the aspect ratio and the DHI mol% for the female hairs decreased, but their plots became continuous with the plots of the male hairs. Additionally, the correlation in the data that combined all males and females increased (*R*^2^ = 0.17, *p* = 0.00020).

The corrected DHI mol% in the female hairs tended to be smaller than that in the male hairs. As the age-dependent plots of the male and female hairs were parallel to each other (Appendix A), the ANCOVA was performed to find a significant sex difference (*p* = 1.2 × 10^−5^). From the fact that the DHI mol% in the male hairs was significantly higher than that in the female hairs, it was thought that the reason why the hair color of the male hairs was darker was due not only to a low pheomelanin mol% but also to a high DHI mol%.

It is known that dopachrome tautomerase or copper ions are involved in the production of DHICA from dopachrome [29,30,31]. The amount or activity of the enzyme or the copper ion in melanocytes should affect the rate of DHICA moieties in eumelanin. It has been shown that Japanese female hairs have a significantly higher copper content than Japanese male hairs [6]. In addition, according to the results of the trace element analyses in hair performed in California [32], the copper contents in female hairs decreased with age whereas its age dependence was small in male hairs. Furthermore, in those with ages under 30, the copper content in female hairs was higher than that in male hairs. These findings, with respect to the age dependence and the sex difference of the copper content in hair, can qualitatively and consistently explain the age dependence and the sex difference in the DHI mol% via the consideration that more copper leads to a higher DHICA (lower DHI) mol% in eumelanin. In order to investigate the causes of the DHI mol% changes with age or sex, it is necessary to examine the melanin composition and the copper content using the hairs from the same subjects. However, since the hair samples used in this study were not necessarily collected from the portion of the hair that is very close to the scalp, the copper content seemed to be affected by the hair washing and hair care processes in daily life, and it was impossible to directly examine the correlation between them.

## 4. Materials and Methods

### 4.1. Hair Samples

All the hair samples (n = 44) used in this study were Japanese male hairs without a history of chemical treatments, such as a perm or hair coloring. They were either provided by volunteers with consent regarding the use of hair samples for this research, or they were donated by volunteers for wigs but could not be used due to being too short for making wigs. The root side portion of the hair fibers was used for the experiments as much as possible. It was unclear which portion was cut before being provided, and the hair samples may be somewhat affected by weathering and repeated shampooing. In the case that the hair samples include some gray hairs, we removed the gray hair fibers and used only pigmented hairs. Hair was washed by immersion in a 2% aqueous solution of polyoxyethylene(9)lauryl ether (Emulgen 109P; Kao Corp., Tokyo, Japan) for 5 min, and then they were rinsed twice with ion-exchanged water and dried in vacuo after replacing with ethanol.

### 4.2. Chemicals

For melanin synthesis and analysis, the following chemicals were used: Tyrosinase (from mushrooms, specific activity of 1715 U/mg) was purchased from Sigma-Aldrich (St. Louis, MO, USA). DHI and DHICA were prepared using the ferricyanide oxidation of 3,4-dihydroxyphenylalanine as described in d’Ischia et al. [33]. Soluene-350 was purchased from PerkinElmer (Waltham, MA, USA). Both 6 M HCl and 57% HI were purchased from FUJIFILM Wako Pure Chemicals (Osaka, Japan). All other commercially available chemicals used were of the purest grade.

For the isolation of melanosomes from hair, the following chemicals purchased from FUJIFILM Wako Pure Chemicals (Osaka, Japan) were used: proteinase K (36 U/mg), papain (reagent grade), protease (42.6 U/mg), and dithiothreitol (for molecular biology). All other commercially available chemicals used were of the purest grade.

### 4.3. Colorimetric Measurement

Hair color was expressed by the parameters *L**, *a**, and *b** in the standard CIEL*a*b* color space defined by the International Commission on Illumination [34]. *L** represented the level of gray from black (*L** = 0) to white (*L** = 100), and *a** and *b** represented the red–green and yellow–blue component, respectively. The positive values of *a** and *b** mean red and yellow color, respectively. We evaluated *L**, *a**, and *b** values for 42 hair samples using a chroma meter (CR-400; Konica Minolta, Tokyo, Japan) with illuminant D65. The measurements were performed on at least five different locations in each hair bundle, and the average values were determined.

### 4.4. Melanin Analyses

The aqueous suspensions of the 44 hair samples were prepared by homogenizing about 15 mg of hair in water at a concentration of 10 mg/mL with a Ten Broeck glass homogenizer. Aliquots of 100 μL (1 mg) were solubilized in Soluene-350 and absorbances at 500 nm and 650 nm for A500 and A650, respectively, were measured [35]. In these analyses, the background absorbances of albino hair (A500 = 0.014, A650 = 0.001) were subtracted. The total melanin amount (TM) was determined by multiplying the A500 value by 101 µg [35].

Hair homogenates were subjected to AHPO [19]. The products, PTCA, PDCA, PTeCA, and TTCA, were analyzed by HPLC under improved conditions using an ion-pair reagent, tetra-*n*-butylammonium bromide [28]. HI reductive hydrolysis followed by HPLC-electrochemical detection was performed to measure 4-AHP levels, as described in Wakamatsu, Ito, and Rees [36]. Then, according to the method reported by Ito et al. [18], aqueous suspensions of 200 μL (2 mg) were subjected to acid hydrolysis with 6M HCl followed by AHPO to measure levels of PTCA, PDCA, PTeCA, and TDCA. It has been found that the TDCA/PDCA ratio has a linear correlation with pheomelanin mol% [18]; we used this relation as a calibration curve to determine pheomelanin mol% in the extracted hair melanin.

### 4.5. Preparation of Synthetic Melanin and Analyses

Mixtures of DHI and DHICA with various mixing ratios were oxidized by tyrosinase to prepare synthetic melanin at pH 6.8 as described in d’Ischia et al. [33]. One hundred microliters of 1 mg/mL suspensions (0.1 mg) of the synthetic melanin were oxidized by AHPO as above to measure PTCA and PDCA levels. With the relationship obtained between the ratio of PDCA/PTCA and the DHI mol% of the mixtures, DHI mol% values in eumelanin of the hair samples were determined [14,37].

### 4.6. Isolation of Melanosomes and Morphology Measurement

We isolated melanosomes from about 0.10 to 0.15 g of hair (n = 44) in mild conditions using three types of enzymes [38]. Next, we determined the morphological parameters of the isolated melanosomes in the same manner as that described for female hairs in the previous paper [14] which is as follows: the dispersions of isolated melanosomes were filtered with suction on a membrane filter with a 0.1 μm pore size (Isopore^®^; Merck KGaA, Darmstadt, Germany), the filtered melanosomes were observed using SEM (JSM-IT500; JEOL, Tokyo, Japan), and digital images were obtained. In order to have a statistical discussion about the size, the major and minor axes of more than 200 melanosomes were measured using Image J software [39] with the digital images of SEM.

### 4.7. Statistical Analyses

Pearson’s product–moment correlation test was used. Simple and multiple linear regression analyses were employed to determine the coefficient of determination *R*^2^ and the *p*-value for the slope coefficient with Excel (Office 365, Microsoft, Redmond, Washington, DC, USA). The ANOVA and ANCOVA were performed with the analysis software R version 4.2.1 [40].

## 5. Conclusions

In this study, we measured hair color, melanin composition, and melanosome morphology in Japanese male hairs of a wide age range; moreover, together with the previous results in the female hairs, we examined their sex differences and factors related to hair color. Additional information about the sex differences in hair was elucidated. Similar to the female hairs, the male hairs tended to darken, and the melanosomes’ size and the TM increased with age. However, the color of the male hairs was significantly darker in younger age periods, and their melanosomes were significantly larger in all ages than in the female hairs. From the analyses using all the data of the male and female hairs, it was found that the factors contributing to hair color were not only the TM but also the mol% of DHI and pheomelanin, suggesting a contribution of these factors in the sex difference of hair color.

## Figures and Tables

**Figure 1 ijms-23-14459-f001:**
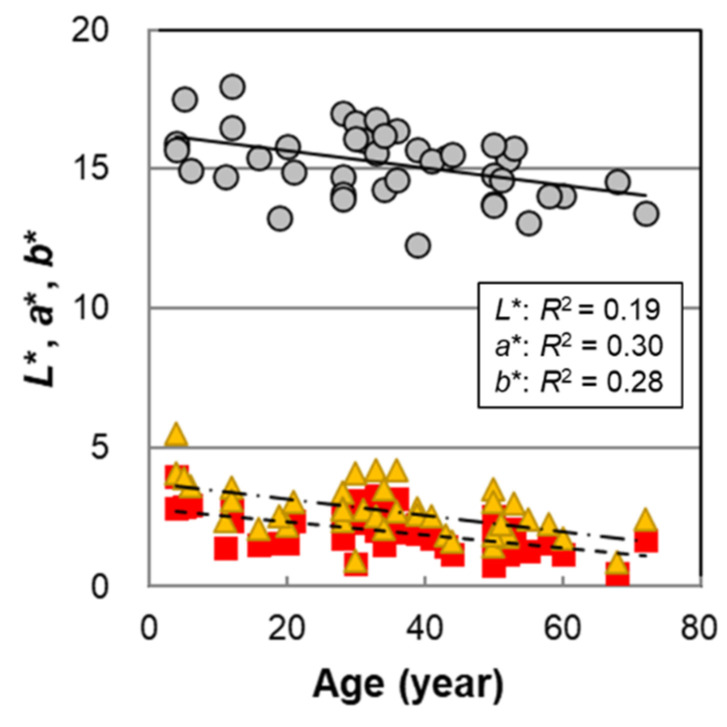
The age dependencies of hair color parameters *L**, *a**, and *b** of Japanese male hair bundles. Circles denote *L**; squares denote *a**; and triangles denote *b**. Lines represent the fitted curves. Solid line denotes *L**; dashed line denotes *a**; and dot-and-dash line denotes *b**. *p* = 0.0035 (*L**), 9.6 × 10^−5^ (*a**), and 0.00020 (*b**).

**Figure 2 ijms-23-14459-f002:**
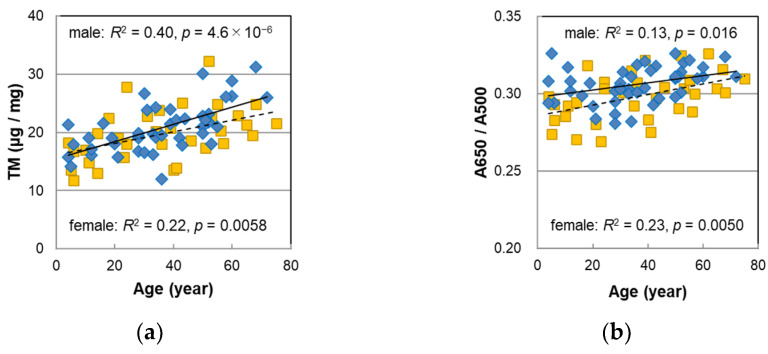
The age dependencies of various values measured with melanin analyses of Japanese male (diamonds) and female (squares) pigmented hairs. (**a**) Total melanin amount (TM). (**b**) Absorbance ratio of 650 nm to 500 nm (A650/A500) of the Soluene-350 solution of hair melanin. (**c**) Level of pyrrole-2,3,5-tricarboxylic acid (PTCA) in hair. (**d**) Level of pyrrole-2,3-dicarboxylic acid (PDCA) in hair. (**e**) Level of pyrrole-2,3,4,5-tetracarboxylic acid (PTeCA) in hair. (**f**) The ratio PTeCA/PDCA. (**g**) 5,6-dihydroxyindole (DHI) mol% in eumelanin of hair. (**h**) Pheomelanin mol% in hair. Lines represent the fitted curves. Solid line denotes male sex and dashed line denotes female sex. The coefficients of determination and *p*-values are shown in each figure.

**Figure 3 ijms-23-14459-f003:**
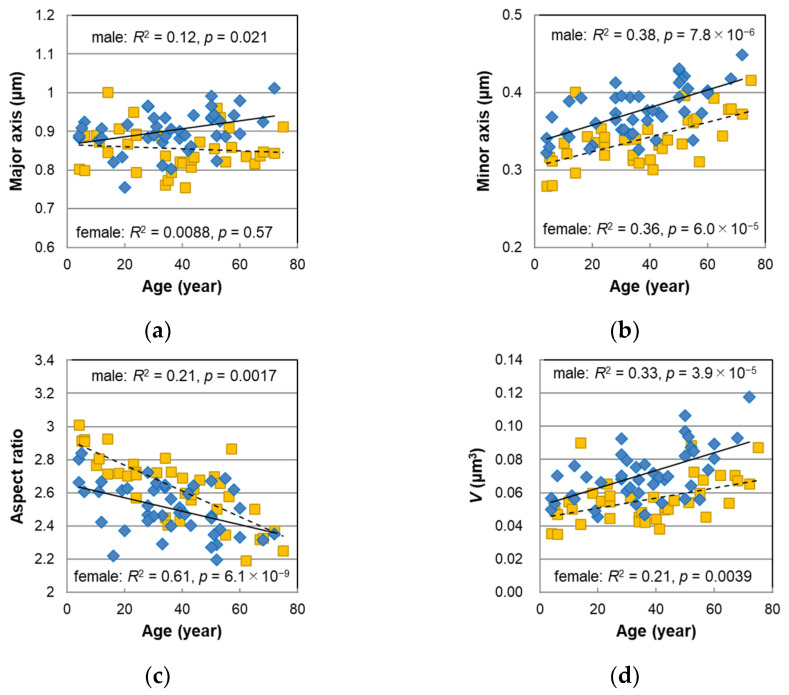
The age dependencies of the morphological parameters determined for the melanosomes isolated from non-chemically treated Japanese male (diamonds) and female (squares) pigmented hairs. (**a**) Mean major axis. (**b**) Mean minor axis. (**c**) Mean aspect ratio. (**d**) Mean volume *V* with an assumption of an ellipsoid. Lines represent the fitted curves. Solid line denotes male sex and dashed line denotes female sex. The coefficients of determination and *p*-values are shown in each figure.

**Figure 4 ijms-23-14459-f004:**
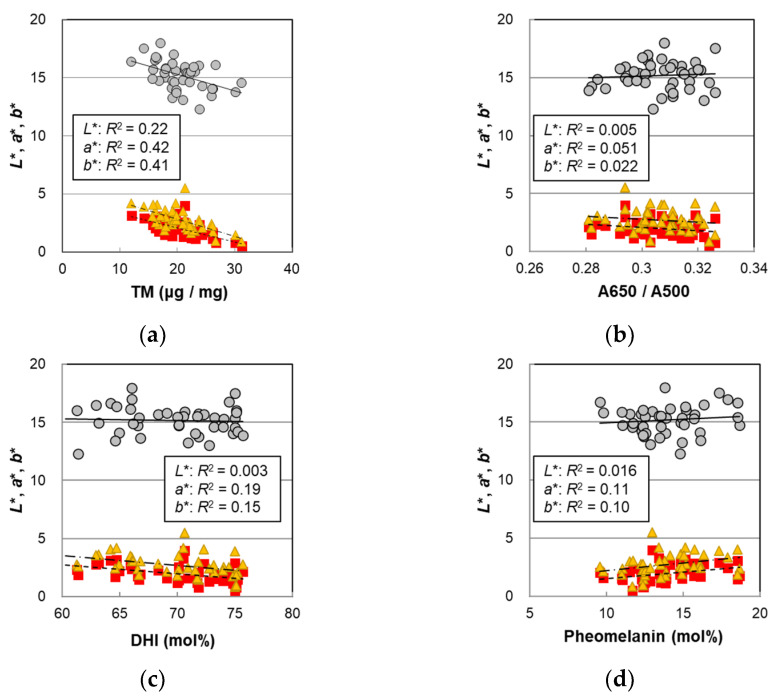
The hair color parameters *L**, *a**, and *b** of the Japanese male hair samples plotted against various values measured with hair melanin analyses. Circles denote *L**; squares denote *a**; and triangles denote *b**. Lines represent the fitted curves. Solid lines denote *L**; dashed lines denote *a**; and dot-and-dash lines denote *b**. (**a**) The color values vs. total melanin amount, *p* = 0.0016 (*L**), 3.0 × 10^−6^ (*a**), and 4.6 × 10^−6^ (*b**); (**b**) the color values vs. A650/A500, *p* = 0.66 (*L**), 0.15 (*a**), and 0.35 (*b**); (**c**) the color values vs. DHI mol%, *p* = 0.74 (*L**), 0.0039 (*a**), and 0.011 (*b**); and (**d**) the color values vs. pheomelanin mol%, *p* = 0.42 (*L**), 0.028 (*a**), and 0.040 (*b**). The coefficients of determination are shown in each figure.

**Figure 5 ijms-23-14459-f005:**
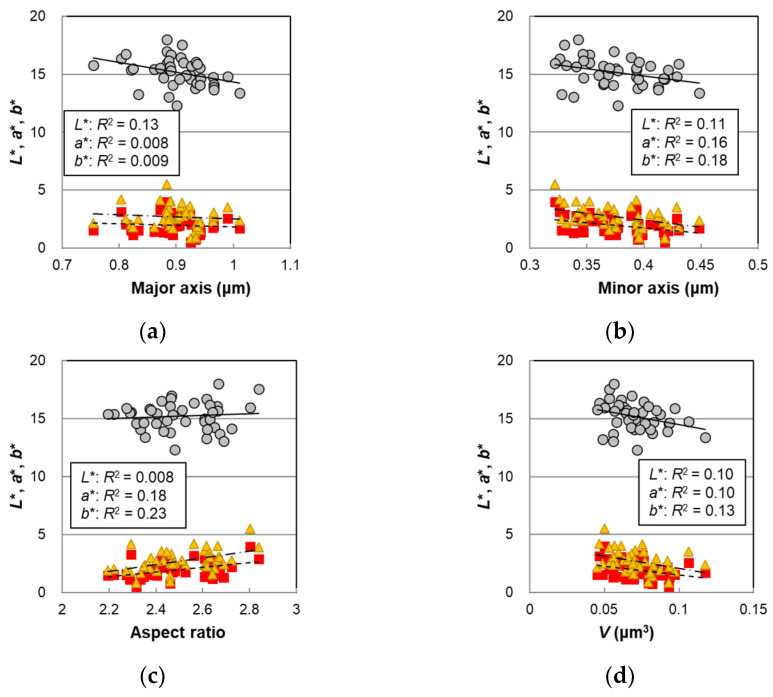
The hair color parameters *L**, *a**, and *b** plotted against morphological parameters determined for the melanosomes isolated from non-chemically treated Japanese male pigmented hairs. Circles denote *L**; squares denote *a**; and triangles denote *b**. Lines represent the fitted curves. Solid lines denote *L**; dashed lines denote *a**; and dot-and-dash lines denote *b**. (**a**) The color values vs. mean major axis, *p* = 0.020 (*L**), 0.58 (*a**), and 0.56 (*b**); (**b**) the color values vs. mean minor axis, *p* = 0.032 (*L**), 0.0099 (*a**), and 0.0048 (*b**); (**c**) the color values vs. mean aspect ratio, *p* = 0.57 (*L**), 0.0050 (*a**), and 0.0013 (*b**); and (**d**) the color values vs. mean volume *V* with an assumption of an ellipsoid, *p* = 0.039 (*L**), 0.037 (*a**), and 0.017 (*b**). The coefficients of determination are shown in each figure.

**Figure 6 ijms-23-14459-f006:**
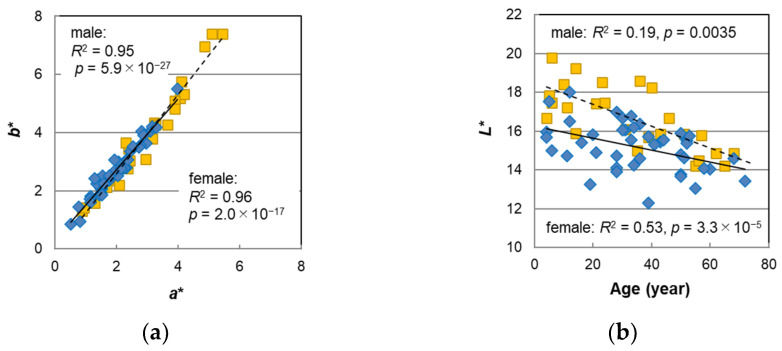
The sex differences of hair color parameters. (**a**) *b** vs. *a**. (**b**) *L** plotted against subject age. Diamond denotes male sex; squares denote female sex. Lines represent the fitted curves. Solid line denotes male sex and dashed line denotes female sex. The coefficients of determination and *p*-values are shown in each figure.

**Figure 7 ijms-23-14459-f007:**
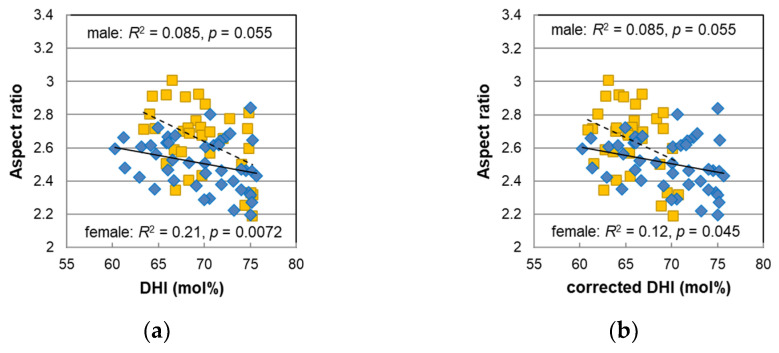
The relationship between the aspect ratio and DHI mol% for male (diamonds) and female (squares) pigmented hairs. The DHI mol% values for female hairs in (**a**) were cited from the original paper [14], and in (**b**), they were corrected assuming that they would be obtained in the improved HPLC condition (see text).

**Table 1 ijms-23-14459-t001:** *p*-values from the regression analysis between hair color parameters and various values.

Parameter		Hair Color		
		*L**	*a**	*b**
Human	Age (year)	0.0035	9.6 × 10^−5^	0.00020
Melanin	Total melanin amount (µg/mg)	0.0016	3.0 × 10^−6^	4.6 × 10^−6^
	A650/A500 ^1^	0.66	0.15	0.35
	DHI ^2^ (mol%)	0.74	0.0039	0.011
	Pheomelanin (mol%)	0.42	0.028	0.040
	PTCA ^3^ level (ng/mg)	0.00041	8.1 × 10^−5^	6.2 × 10^−5^
	PDCA ^4^ level (ng/mg)	0.0088	2.2 × 10^−7^	6.2 × 10^−7^
	PTeCA ^5^ level (ng/mg)	0.0045	0.082	0.038
Melanosome	Major axis (µm)	0.020	0.58	0.56
	Minor axis (µm)	0.032	0.0099	0.0048
	Aspect ratio	0.57	0.0050	0.0013
	Volume (µm^3^)	0.039	0.037	0.017

^1^ The ratio of absorbance at the wavelength of 650 nm (A650) to 500 nm (A500), ^2^ 5,6-dihydroxyindole, ^3^ pyrrole-2,3,5-tricarboxylic acid, ^4^ pyrrole-2,3-dicarboxylic acid, and ^5^ pyrrole-2,3,4,5-tetracarboxylic acid.

**Table 2 ijms-23-14459-t002:** Sex differences for melanin and melanosome parameters measured with ANCOVA or ANOVA.

Parameter		Parallelism	Statistic	*p*-Value
Melanin	TM	No	ANOVA	0.30
	A650/A500	Yes	ANCOVA	0.0058
	DHI mol%	Yes	ANCOVA	0.96
	Pheomelanin mol%	Yes	ANCOVA	5.3 × 10^−7^
Melanosome	Major axis	No	ANOVA	0.00024
	Minor axis	Yes	ANCOVA	6.0 × 10^−9^
	Aspect ratio	No	ANOVA	0.0019
	Volume	Yes	ANCOVA	7.1 × 10^−7^

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
