# Peer review of "Effects of Aging on Hair Color, Melanosomes, and Melanin Composition in Japanese Males and Their Sex Differences"

_ijms, 2022, doi:10.3390/ijms232214459_

Round 1
Reviewer 1 Report
Overall this study appears well done but with the methods relegated toward the end the reader is forced to jump around to figure out what the authors are doing and if it is a reasonably scientific study. I'd suggest some minor improvement in English grammar for commas, singular vs plural use of words, and cut down on some of the run-on sentences. Several statements are too definitive and should be moderated or better cited (beyond just your own previous research) and explained so you don't have to go to those studies to understand this paper.
Some specific comments:
The introductory sentence is run on and using the phrase "attractive appearance" brings a level of subjectivity that is not necessary for this paper - cut the word attractive and just state it has an impact on a person's appearance
line 37 "Before graying, the color of pigmented hair changes with age" is too definitive of a statement - needs multiple citations from longitudinal studies (not just your own work) on multiple population groups to support this very broad statement, otherwise I suggest changing to "can" or "may" change with age or specifying Japanese hairs from females and add citation 14 though then you run into issues of over citing your own research
line 111 " Representative images of the isolated melanosomes" here you need to indicate SEM images and that the melanosomes were chemically isolated from the hairs. This helps the reader to understand some of the methods used prior to getting to the methods section since that is later in this paper
Scale bars should be in all photomicrographs - not having them in should be explained otherwise it could be interpreted as the images were cropped or otherwise altered.
methods - the word "part" in lines 396 and 397 should be changed to portion
line 407 "as described in d’Ischia et al. (33)" while this is often done it makes it harder for the reader to know what you did based on just this paper and for those who have paywall issues it would help if you elaborated on the method as well as indicating it followed what was previously done - same comment for other "as described" statement (line 434) "in same manner" (line 450) in methods section
conclusions line 466 " as a result, new knowledge about sex differences in hair has been found" is awkward phrasing. Suggest changing to Additional information about sex differences in hair has been elucidated (or explained).
Author Response
Dear Reviewer 1,
Thank you for your kind review. We have revised our manuscript according to your comments. Please find the attached file.
Takashi Itou

Reviewer 2 Report
The article "Effects of Aging on Hair Color, Melanosome, and Melanin Composition in Japanese Males and Their Sex Difference" is a continuation of previous research done on a group of women. The author described the topic in an interesting way. Only I would ask for an emphasis on the number of samples used in each analysis. I have no further comments. I believe that the article is suitable for publication in IJMS.
Author Response
Dear Reviewer 2,
Thank you for your kind review. We have revised our manuscript according to your comments. Please check the attached file.
Takashi Itou
